# Corrosive Effect of Wood Ash Produced by Biomass Combustion on Refractory Materials in a Binary Al–Si System

**DOI:** 10.3390/ma15165796

**Published:** 2022-08-22

**Authors:** Hana Ovčačíková, Marek Velička, Jozef Vlček, Michaela Topinková, Miroslava Klárová, Jiří Burda

**Affiliations:** Department of Thermal Engineering, Faculty of Materials Science and Technology, VSB-Technical University of Ostrava, 17. listopadu 2172/15, 708 00 Ostrava, Czech Republic

**Keywords:** corrosion, refractory, biomass, thermal processing, wood ash

## Abstract

In terms of its chemical composition, biomass is a very complex type of fuel. Its combustion leads to the formation of materials such as alkaline ash and gases, and there is evidence of the corrosive effect this process has on refractory linings, thus shortening the service life of the combustion unit. This frequently encountered process is known as “alkaline oxidative bursting”. Corrosion is very complex, and it has not been completely described yet. Alkaline corrosion is the most common cause of furnace-lining degradation in aggregates that burn biomass. This article deals with an experiment investigating the corrosion resistance of 2 types of refractory materials in the Al_2_O_3_-SiO_2_ binary system, for the following compositions: I. (53 wt.% SiO_2_/42 wt.% Al_2_O_3_) and II. (28 wt.% SiO_2_/46 wt.% Al_2_O_3_/12 wt.% SiC). These were exposed to seven types of ash obtained from one biomass combustion company in the Czech Republic. The chemical composition of the ash is a good indicator of the problematic nature of a type of biomass. The ashes were analyzed by X-ray diffraction and X-ray fluorescence. Analysis confirmed that ash composition varies. The experiment also included the calculation of the so-called “slagging/fouling index” (I/C, TA, Sr, B/A, Fu, etc.), which can be used to estimate the probability of slag formation in combustion units. The corrosive effect on refractory materials was evaluated according to the norm ČSN P CEN/TS 15418, and a static corrosion test was used to investigate sample corrosion.

## 1. Introduction

Worldwide, 80% of electricity is produced using fossil fuels. According to the International Energy Agency (IEA), electricity production reached approximately 25.8 T-kWh in 2020, and an increase to 36.5 T-kWh is expected by 2040 [1]. As stated by the World Bioenergy Association, 59.2 NPP/year, i.e., 10.3% of the global energy supply, comes from biomass [2]. Biomass is becoming a popular source of energy which can be used in various ways. Electricity produced from biomass currently corresponds to 493 TWh, which is approximately 2% of the world’s electricity production [2]. Using biomass as a raw material for powerplants is certainly interesting and useful; however, this technology also has certain disadvantages. The use of biomass in powerplants leads to the formation of residue, called biomass ash. It is estimated that around 480 million tons of ash are produced every year by biomass powerplants worldwide. This is similar to coal ash, with 780 million tons per year [2].

The most frequently burned material is wood (64%), followed by cereals and plant residue from agricultural production. In general, it can be said that the average percentage of ash produced by burning biomass ranges between 1 and 6%; for wood, it is 0.6–1.6%; for bark, it rarely exceeds 3%; straw produces an ash content of around 5%, while grass produces 7%. At the other end of the spectrum, the ash content produced by black coal is significantly higher, reaching 20–30%, and from brown coal, this amount can be even greater [3]. Ash represents a variable composition of mineral and inorganic components.

During the combustion process, ash continuously changes its physical and chemical properties, the final product being a molten mixture of original minerals, various eutectics, and elements. Ash causes various problems, especially corrosion, erosion, stickers, etc. [4] If the melting temperature of ash during combustion is t_ash_ < t_flame_, then the grate of the hearth can become clogged. Ash layers on the walls of the furnace diffuse into the lining, which then peels off in thin layers. The combustion chamber of the boiler must therefore be structurally adjusted such that the flame temperature drops below the ash melting temperature, i.e., the temperature on the grate should be lower than the melting temperature of the biomass ash [5].

The major problem from a chemical point of view is corrosion, which comes from the interaction between a refractory and a corrosive medium: gas, molten metals, molten glass, molten salts, or slag. It results in a loss of mass and thickness and in the degradation of the material properties [6]. The corrosion of refractory materials is a combination of external and internal physical and chemical influences.

The process is basically a chemical reaction between the refractory material and the slag or metal. Reactants are transported to the interface of the refractory material, and, in turn, the product reacts and is transported to the liquid phase. The dissolution of refractory materials in the melt is controlled by diffusion. Three types of corrosion have been defined: surface, dimple, and undersurface corrosion [7].

Alkaline corrosion, or “alkaline oxidative bursting”, is extremely common, effective, and particularly harmful to alumina–silicon (Al–Si) lining systems, and it is usually observed in the temperature range of aggregates of 800–1000 °C. During biomass combustion, damage to the refractory lining is observed (Figure 1) as the peeling of surface layers, cracking, the bending of individual parts of the lining, the bulging of entire walls, and eventually their collapse [8,9].

The main difference between coal ash and biomass ash is that coal ash contains higher amounts of SiO_2_ and Al_2_O_3_, but it contains lower amounts of K_2_O and Na_2_O. The eutectic of Al–Si forming fly ash lies above 1200 °C, while the eutectic of plant fly ash is much lower. Eutectic temperatures for mixtures of alkali metals together with silica or phosphorus have a low melting point: Na_2_O.2SiO_2_ (874 °C), K_2_O.4SiO_2_ (770 °C), and 2CaO.3P_2_O_5_ (774 °C) [10].

Aluminosilicate refractories are based on the SiO_2_-Al_2_O_3_ system. The equilibrium diagram of this system is given in Figure 2, marking various refractories. The main phase in the Al–Si binary diagram is mullite (3Al_2_O_3_.2SiO_2_) [11], which increases the resistance of the refractory material against the corrosive effects of ash [11].

In the AL–SI system, new phases are often formed as a result of different chemical reactions, gradually degrading the system. The newly formed products have a larger volume than the original material, with expansion being reported between 7 and 30%. This creates compounds in the lining or on its surface that have chemical compositions and physical parameters different from the lining itself [12]. Ternary diagrams of the types Na_2_O-Al_2_O_3_-SiO_2_ and K_2_O-Al_2_O_3_-SiO_2_ also describe the formation of individual phases in the given system (see Figure 3).

The corrosion mechanism in the Na–Al–Si system includes the formation of albite (NaAlSi_3_O_8_), nosean (Na_8_Al_6_Si_6_O_28_S) [15,16], and natrosilite (Na_2_Si_2_O_5_) by Equation (1), which further reacts with mullite (Al_6_Si_2_O_13_) to form albite (NaAlSi_3_O_8_) and aluminum oxide according to Equation (2). Nepheline (NaAlSiO_4_) can also be formed according to Equation (3). Nosean is rarely reported in the literature as a corrosion product. However, due to its structural similarity to nepheline, it can also be expected to produce swelling. The reaction can be described by Equation (4) [17]:Na_2_SO_4_ + 2SiO_2_ = Na_2_Si_2_O_5_ + SO_2_ + 1/2O_2_(1)
Na_2_Si_2_O_5_ + 2Al_6_Si_2_O_13_ = 2NaAlSi_3_O_8_ + 5Al_2_O_3_(2)
2NaAlSi_3_O_8_ + Al_6_Si_2_O_13_ + 3Na_2_SO_4_ = 8NaAlSiO_4_ + 3SO_2_ + 3/2O_2_(3)
4Na_2_SO_4_ + 3Al_6_Si_2_O_13_ = Na_8_Al_6_Si_6_O_28_S + 6Al_2_O_3_ + 3SO_2_ + 3/2O_2_(4)

In the case of high-alumina refractories (>45% Al_2_O_3_) containing mullite (A_3_S_2_) and cristobalite (SiO_2_), reaction with NaO_2_ above 1000 °C forms nepheline (NaS_2_) and α-Al_2_O_3_ according to Equation (5). As can be deduced from the ternary diagram K_2_O-Al_2_O_3_-SiO_2_, at a lower content of Al_2_O_3_ < 30%, orthoclase KAS_6_ is formed, and at a content of Al_2_O_3_ > 30%, new phases of leucite (KAS_4_) are formed according to Equation (6):3Al_2_O_3_·2SiO_2_ + Na_2_O → Na_2_O·Al_2_O_3_·2SiO_2_ + 2Al_2_O_3_(5)
K_2_O·Al_2_O_3_·6SiO_2_ → K_2_O·Al_2_O_3_·4SiO_2_ + 2SiO_2_(6)

Since the composition of biomass ash encourages the formation of eutectic melts, it is advisable to use high-alumina refractory materials with an Al_2_O_3_ content > 80% or to add silicon carbide for these linings. The compound, aluminosilicate-based materials mainly include products containing oxide-less constituents—graphite and silicon carbide

Materials in Al_2_O_3_-SiO_2_-SiC systems combine the high thermal conductivity and chemical inertness of silicon carbide with the chemical and thermal stability of aluminosilicate and corundum. The products are therefore highly resistant to corrosion by liquid metals, as well as to sudden changes in temperature.

SiC oxidizes according to Equations (7) and (8) and creates an amorphous SiO_2_ film on the surface [18,19]:SiC + 1.5O_2_ → SiO_2_ + CO(7)
SiC + 2O_2_ → SiO_2_ + CO_2_(8)

To prevent graphite oxidation, firing is carried out without any contact between the fired product and oxygen. The firing temperature is chosen to create a ceramic bond in the products. At present, the process of quick firing is used, ensuring a reducing atmosphere in the kilns at higher temperatures and while cooling the products.

K_2_O and Na_2_O, in the form of alkaline vapors, are capable of diffusing into the refractory matrix, and then they react with Al_2_O_3_ and SiO_2_ components to form K-aluminosilicate and Na-aluminosilicate phases [20]. In the AL–SI binary system, potassium pairs react according to Equations (9)–(11). The most harmful is the presence of free SiO_2_ and Na_2_O, which increase the reaction rate at high temperatures and support the formation of reactive glassy phases, according to Equation (12) [20]:K_2_O + SiO_2_ → K_2_O.SiO_2_(9)
3 (K_2_O.2SiO_2_) + 3Al_2_O_3_.2SiO_2_ → 3 (K_2_O Al_2_O_3_.2SiO_2_) + 2SiO_2_(10)
K_2_O.Al_2_O_3_.2SiO_2_ + 2SiO_2_ → K_2_O.Al_2_O_3_.4SiO_2_(11)
2SiO_2_ + Na_2_O → Na_2_O·2SiO_2_(12)

The so-called slagging/fouling index can be used to estimate the probability of slag formation in combustion units during biomass combustion. Slagging/fouling means the formation of layers (sticky, melted, or soft) of ash particles on heat exchange surfaces. A summary of slagging and fouling indices and their calculation are presented in Table 1.

The *SiO_2_ index* is often the predominant element in biomass samples and causes the formation of melt, or “stickers”, therefore giving it the characteristic of being slag-forming.

The *chlorine index Cl* acts as an accelerator of the reaction between K and SiO_2_, which leads to the formation of fused glass deposits and the formation of slag at boiler operating temperatures of 800–900 °C [23].

Ash-deposition potential may be evaluated in terms of base-to-acid (B/A). The *basicity index B/A* (base/acid ratio) is based on the general rule that basic oxide compounds lower the melting point, and acidic compounds raise it. The B/A ratio is an indication of the fusion and slagging potential of ash. *I/C* (iron/calcium ratio) stands for Fe_2_O_3_/CaO, e.g., ash with a ratio of Fe_2_O_3_/CaO = 0.3/3.0 containing eutectics that increase slag formation.

The *Fouling index Fu* (fouling index) is the B/A ratio, also taking into account the alkali content (Na_2_O + K_2_O). Fouling refers to the dry deposition of ash particles or the condensation of volatile inorganic components on heat transfer surfaces. The normal percentage of alkali in biomass ash is between 25 and 35%, and it forms a eutectic in combination with silica.

Ash has a high viscosity (Sr) value, *slag viscosity index Sr* [24], so it will have a low tendency to slag. The *TA (total alkali) index* assesses the fuel’s ability to form ash layers. Values of individual ash samples, defined based on the above-mentioned indices, are summarized in Section 3.2.

The chemical composition of ash is a good indicator of the problematic nature of biomass. For biomass fuels, massive slagging of heat exchange surfaces of boilers occurs during combustion. Ash composition and atmosphere in a combustion chamber influence the ash-melting temperature [10]. Indicators tell us of the characteristics of ash in terms of their influence on the formation of the glassy phase, and thus their tendency to slag and clog linings, heat exchange surfaces, and gas flow routes. These indices are based on chemical composition of biomass and its combustion. The equations are mainly based on fuel evaluation. However, since there is no specific index for biomass, it is possible to apply these indices to this type of fuel as well.

## 2. Materials and Methods

### 2.1. Ashes from Wood Biomass Combustion 

Seven different types of ash from different types of wood biomass were used for the experimental portion of our study. All of these were obtained from the Czech Republic, mainly from the Moravian–Silesian Region, but one was from the Central Bohemian Region. Ashes utilized during the experimental portion were used in the original form for the crucible test. the granulometry was not adjusted. More information about the ash samples is presented in Table 2. 

### 2.2. Refractory Materials

Tested refractory materials were manufactured by one of the largest producers and suppliers of refractory products and raw materials in the Czech Republic. Two types of shaped refractory materials, belonging to the silica–aluminum group, were selected for the corrosion experiment.

The first type was quality labeled as STV. It is a shaped refractory material classified as standard fire clay. The second type was quality labeled as ARS60N and is classified high alumina. The parameters of the mentioned tested materials with their properties are shown in Table 3.

### 2.3. Corrosion Crucible Test and Evaluation Method

The crucible test gives only approximate results. The refractory cube was filled with corrodent and heated to the testing temperature for a specified period. The testing conditions (temperature and corrodent composition) may reflect the expected service conditions, but in some situations, a more aggressive corrodent and/or high temperature may be used to speed up the attack to determine the resistance of the refractory to the corrosive liquid in a relatively short time. The crucible test is described step by step in Figure 4. The refractory cuboid sample with a cylindrical hole in the central portion was filled with corrosive, medium/powdered ash with a heating temperature of 1200 °C for 2 h. After cooling, the tested sample was cut through along the vertical axis, and the corroded portion was measured.

After the corrosion test, samples were visually checked for compactness, potential cracks, and holes in the sample and walls. The ČSN P CEN/TS 15418 method [25] and the internal regulation method of P-D Refractories CZ a.s. [26] were used for test evaluation.

The classification used for reporting the condition of the crucible with defined parameters [25] U: unaffected/no visible attack; LA: lightly attacked/minor attack; A: attacked/clearly attacked and C: corroded/completely corroded. In addition to the above-mentioned evaluation regulations, another internal regulation method of P-D Refractories CZ was also used [26]. 

Table 4 shows the parameters of the classification after the corrosion test. Two evaluation methods may sometimes be requested by a customer or company testing laboratory, and the parameters can be used for comparison.

### 2.4. Characterization Methods

The chemical composition (XRF) of the ash was determined by energy-dispersive X-ray fluorescence spectroscopy (ED-XRF) on the SPECTRO XEPOS (Spectro Analytical Instruments, Kleve, Germany). Powdered samples were shaped/pressed into tablets for XRD measurement.

The mineralogical composition (XRPD) of the samples was evaluated using X-ray diffraction analysis on the X-ray diffractometer MiniFlex 600 (Rigaku, Tokyo, Japan) equipped with a 0Co tube and a D/teX Ultra 250 detector. XRD patterns were recorded in a 5–90° 2θ range with a scanning rate of 5° min^−1^.

## 3. Results and Discussion

### 3.1. Ash Characterization

Chemical analysis is a good indicator for determining the problematic nature of biomass. The chemical composition of all of the ash types is presented in Figure 5. Biomass ash almost always contains carbonates, especially calcite, and very often portlandite, as well as a proportion of organic carbon.

Oxides in biomass ash can be divided into acidic (SiO_2_, Al_2_O_3_, TiO_2_, etc.) and basic (K_2_O, CaO, MgO, Na_2_O, Fe_2_O_3_, P_2_O_5_, etc.). Acidic oxides increase the melting point of ash. The higher the content of acidic oxides, the higher the melting point. On the other hand, basic oxides lower the melting point of the ash.

The predominant oxides are SiO_2_ and CaO. A high level of CaO is typical for wood. The higher the content of basic oxides, the lower the melting point. SiO_2_ plays an important role as a glass-forming oxide, while CaO and K_2_O reduce the viscosity of the resulting glass-forming melt. The nature of the oxides and their representation determines the formation of other compounds and the behavior of the refractory material in contact with the corrosive agent. Ash was analyzed by XRDF, and this showed variable sample composition. The percentage of single oxides is as follows: SiO_2_ 9.13–55.17 wt.%, CaO 16.33–41.79 wt.%, Al_2_O_3_ 0.98–10.14 wt.%, Fe_2_O_3_ 1.80–13.16 wt.%. For alkali oxides it is Na_2_O 0.38–12.23 wt.% and K_2_O 6.11–19.17 wt.%. The amount of Cl is around 0.6 wt.%.

In terms of chemical composition, ash resembles low-melting glass. The variability of chemical composition complicates accurate representation in a ternary diagram. An approximate composition based on the largest content of wt.% of oxides is shown in the diagram. Four ash types, labeled P_020_, P_033_, P_059_ and P_060_, are marked in the CaO-Al_2_O_3_-SiO_2_ ternary diagram, and two types, labeled P_031_ and P_019_ are marked in the K_2_O-SiO_2_-CaO system, as presented in Figure 6.

The next method of ash characterization was X-ray powder diffraction phase analysis (XRPD). The samples were compared to the reference diffractogram database published by ICDD (PDF-2) in the range of 5–90° 2theta. The results for the analyzed samples are presented in Table 5, where there is an overview of the phases in the samples.

As confirmed by the analysis, the most frequently recurring phases are quartz, anorthite, calcium silicate, hematite, anhydrite, and microcline. In ash samples P_059_ and P_060_, there were seven phases identified as portlandite; microcline, leucite, and portlandite occur in both. Samples P_019_ and P_032_, were especially rich in the glass phase.

### 3.2. Calculation of the Slagging and Fouling Indices 

To predict slagging/fouling in a combustion furnace, it is possible to use indices for the SiO_2_, basic/acid ratio, silica/alumina ratio, fouling, iron/calcium ratio, and total alkalis, as summarized in Table 6. A special index only for biomass does not exist, but many authors have calculated these indices with regard to the probability of slag forming in combustion units.

In the case of SiO_2_ content in P_020_, P_033_, P_059_ and P_060,_ they have a high inclination towards slagging. The high levels of silica in wood biomass ashes may have been caused by contamination with different elements (clay, sand, etc.); also, each part of the wood plant may contain different amounts of oxides. According to chloride content, extremely high fouling inclinations were observed in samples P_032_ = 1.74 and P_020_ = 0.47, while a low fouling inclination with a value > 0.2 was calculated for P_033_ = 0.1. 

Index B/A confirmed that all wood biomass ash samples included in this study showed high to extremely high values of slagging (>1), as shown in Table 6 and Figure 7. The highest value B/A = 8.1 was for ash P_059_, which will have a tendency toward slag formation. The low B/A index for P_033_ ash can be attributed to its high SiO_2_ content, which implies an increased presence of acidic compounds. No ash samples had a B/A value lower than 0.5. Samples which had the highest B/A index and a high Na_2_O content showed an inclination towards fouling.

The I/C value for the tested samples was >0.3 and represents a small risk for the degradation of refractory materials. S/A index values were higher for all tested ash samples. the slagging tendency based on the silica/alumina ratio (S/A) was high for all the ash samples because no value was lower than 3 (Figure 7).

As a final result, higher Fu values correspond to higher fouling tendencies. The extreme value of the fouling index (Fu) was observed in sample P_019_, with Fu = 240.1, and P_031_ Fu = 72.1, as presented in Figure 8. Fouling index values over 40 indicate a high tendency to fouling. The fouling tendency based on total alkalis was high for the types of ash mentioned below.

The value of total alkali content (TA) was never less than 0.4. (Figure 8). The lowest TA index was observed in P_060_ TA = 5.3, and it was 6 times higher in P_019_ = 29.7. The maximum value was 0.4, and there was a higher tendency towards the fouling of refractory materials during biomass combustion. Slagging probability was high for ash samples based on their Sr ratios, which were lower than 65. In this study, the calculated index for Sr corresponded to five ash types. Samples P_020_ and P_033_ had the middle value between 65 and 72. Based on calculations, the ash will tend to form slag and deposits.

### 3.3. Corrosion of Refractory Materials 

Two types of refractory materials were selected for this portion of the corrosion experiment, STV fire clay (containing 53 wt.% SiO_2_, 42 wt.% Al_2_O_3_), shown in Figure 9, and ARS60N high alumina (containing 28 wt.% SiO_2_, 46 wt.% Al_2_O_3_, and 13 wt.% SiC), shown in Figure 10. Refractory materials underwent a crucible test under the following conditions: 2.9 g of ash as corrosion agent/2 h on maximum temperature at 1200 °C. All ash types melted at the suggested testing temperature.

Resistance to eutectic melts was determined by the refractory quality of STV at 1200 °C. As shown in Figure 9, infiltration was relatively low in samples P_019_ and P_060_ with refractory materials, whereas the attack of ashes P_032_ a P_031_ achieved a high intensity with STV refractory materials. 

The evaluation of the corrosion attack by two methods is presented in Table 7. The corroding interaction was most intensive between P_032_ and STV refractory surface. P_032_ had the highest proportion, 1.77 wt.% Cl, a high content of alkalis (Na_2_O + K_2_O), namely 16 wt.%, a high CaO content of 26 wt.%, and 19 wt.% SiO_2_. B/A = 2.3 ^ex^ was extreme, and the ash can be defined as basic. The slag viscosity index was Sr = 35.2 ^h^ and index Fu = 46.6 ^h^ was high, too. The ash showed a tendency towards slagging, and CaO and K_2_O reduced the viscosity of the slag. Basic oxides lowered the ash melting point.

Cl acts as an accelerator of reactions between K_2_O and SiO_2_, which leads to the formation of fused glass deposits and the formation of slag at the boiler operating temperature of 800–900 °C [23]. According to Table 7 and normative regulation [25], the reaction between the STV quality refractory and ash P_032_ was evaluated as attacked/clearly attacked, and according to [26], the distinctive attack is >7 mm corrosion and/or infiltration/slight cracks.

P_031_ attack had an effect in contact with refractory materials that was very similar to that of P_032_. All index values were categorized as high, but the amount of Cl was only 0.16 wt.%, unlike that of P_031_. The crucible test of STV corrosion by P_031_ was evaluated according to [25] as attacked/clearly attacked and [26] slight attack/<6 mm corrosion and/or infiltration/no cracks.

From the visual point of view (see Figure 9), the sample tested by P_033_ formed melt, which represents a low-melt eutectic compound, because K_2_SiO_3_ and Na_2_SiO_3_ present low melting temperatures. P_033_ contained 55.1 wt.% SiO_2_, alkali (Na_2_O + K_2_O) 7 wt.%, and only 0.1 wt.% Cl. Index B/A was only 0.5, but a high index Sr = 71.7, which had a negative effect to refractory materials. A similar effect was observed on samples attacked by P_020_. STV refractory materials, after the application of P_019_ and P_060_, may appear to be inert to ash, but a little infiltration was noticeable upon closer examination. Nevertheless, the evaluation according to the methodology was uniform.

An increase in temperature can be a significant indicator of changes in refractory materials. A problem may occur if temperature fluctuations occur–in this case, the combustion temperature could be 200 °C higher than the normal operating temperature [29], and the corrosion of refractory materials can be very intensive.

The formation of slag in the furnace depends on the chemical and mineralogical composition of the ash and on the conditions in the furnace (temperature, reduction, or oxidation zone, etc.). When fuel is burnt, combustion produces ash. In this case, residue will always form on the walls of the hearth and the heat exchange surfaces [18]. Slag deposits can be further divided into so-called “sintered deposits”. (They are formed by dust particles trapped in a liquid or plastic state and stick to the wall.) The second type is hard deposits having a layered structure. The viscosity of the slag depends on the composition of the ash [13,14,30].

The second tested sample was high-alumina refractory ARS60N with a content of 46 wt.% Al_2_O_3_, 28 wt.% SiO_2_, and 13 wt.% SiC (Figure 10). From the visual point of view, the refractory material was resistant to ash attack at 1200 °C. None of the samples showed total corrosion leading to disintegration. P_032_ showed negative effects on the refractory material ARS60N, just as it did with the STV quality. P_032_ caused slight changes on the surface structure with very fine cracks. The infiltration of ash was 6.4 mm, and in the structure of the tested refractory materials, it formed spaces without grains, which changed to compact slag in the middle of the sample. However, the integrity of the samples was preserved. A detailed photo of the changing structure of P_032_ is presented under the main pictures. As already mentioned, this ash has a tendency towards slagging. A very similar effect was observed in the middle of the refractory material, which formed after the attack of ash P_033_. According to Table 8 and the literature [25], the corrosion effect between refractory quality ARSN60 and P_032_ described as attacked/clearly attacked, and according to Plibrico [26], it presented as a distinctive attack />7 mm corrosion and/or infiltration/slight cracks.

The viscosity index of P_032_ is Sr = 35.2 ^h^, and that of P_031_ Sr = 25.3 ^h^. According to [24], ash with a value (<65) viscosity Sr will have a high tendency towards slagging. On the other hand, P_020_, with index Sr = 67 ^m^ and P_033_ Sr = 71.7 ^m^, will display the opposite trend. The effect of P_020_ is visually comparable for both STV and ARS60N refractory materials. The interaction between P_020_ and ARS60N samples formed a very slight layer of glassy character with an infiltration of about 6–8 mm. In terms of chemical composition, P_020_ contained 46 wt.% SiO_2_, 16 wt.% CaO, and 11 wt.% alkali oxides (Na_2_O + K_2_O). This ash is acidic, as the ratio is B/A = 0.7 ^m^. TA = 11.5 ^m^ and Fu = 7.7 ^m^ [24], so this ash should have a low tendency towards slagging. Other tested ash types showed invisible or minimum attack with refractory materials. ARS60N presented a higher level of corrosion resistance against the tested ash types than to STV. This portion of the experiment showed that the suggested temperature of 1200 °C during the corrosion crucible test should not be exceeded for tested STV and ARS60N refractory materials with the above-mentioned tested wood ash types.

## 4. Conclusions

STV quality refractory materials contain 53 wt.% SiO_2_ and 42 wt.% Al_2_O_3_), while in ARS60N, the quantity of oxides is 46 wt.% Al_2_O_3_, 28 wt.% SiO_2_, and 13 wt.% SiC). These materials belong to the alumina–silica system and were tested for their corrosion resistance. The results obtained within this research can be summarized as follow:Seven types of wood ash were used as corrosive media in their original forms. All types of ash were obtained by biomass combustion. The testing method was a crucible test at 1200 °C.The intensity of the corrosive effect was determined by two regulations. Neither a corrosion test nor visual evaluation revealed a greater penetration or destruction of the samples and that was not recorded, but only light/middle corrosion. From all tested ash samples, the one labeled P_032_ achieved the highest corrosive effect on both tested alumina–silica refractory materials.A less-intensive corrosion attack was observed in P_031_ with STV and ARS60N with P_020_. Visible signs of corrosion included melts, fine cracks, infiltration, and in some cases, structural changes. The above-mentioned results suggest that the poorer the material is in silicon dioxide, the better it resists corrosion.The chemical stability of the phases increases as follows: amorphous silica (SiO_2_) < cristobalite (SiO_2_) < quartz (SiO_2_) < andalusite (Al_2_SiO_5_) < mullite (Al_6_Si_2_O_13_) < corundum (Al_2_O_3_).Upon a visual check, fire clay materials with SiO_2_ > Al_2_O_3_ content presented a higher infiltration of the corrosion medium into the sample with an apparent porosity similar to that of sillimanite refractory materials.This may lead us to the conclusion that a higher level of Al_2_O_3_ content can increase corrosion resistance, as is often reported. An effective solution is also a partial replacement of the oxide with silicon carbide SiC, which has shown a favorable effect on the resistance of refractory materials. A high refractory-material-testing temperature can be a good indicator of a possible corrosive effect on the tested refractory material if there is undesirable temperature fluctuation during combustion.The nature and behavior of biomass ash can be defined by calculating slagging/fouling indices (B/A, Sr, Fu, TA, etc.), which are applied to the calculations of conventional fuels.Based on the chemical composition and calculated indices for wood biomass ash samples, it was confirmed that many of the values are above the limits and should tend to form slag and fouling. However, not all samples confirmed this prediction.

## Figures and Tables

**Figure 1 materials-15-05796-f001:**
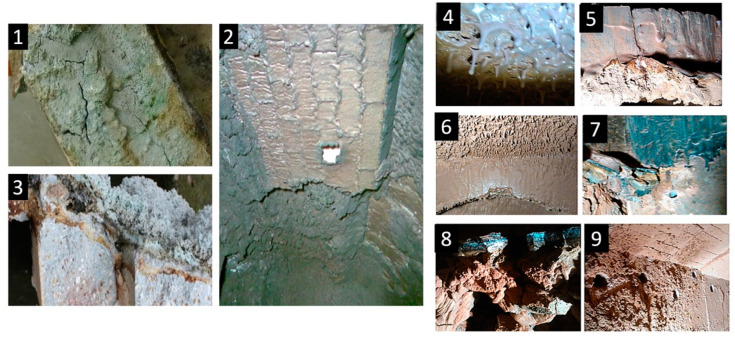
Degradation of refractory materials in boilers after combustion of different types of biomasses [8,9]: (**1**) the damage to refractory materials after 1 year of the combustion of wood chips; (**2**) the furnace vault after 1.5 years of the combustion of chipboard; (**3**) the corroded part of refractory samples after 2 years of combustion of plant biomass, and (**4**–**9**) the presentation of the defects of the refractory lining after the combustion of biomass for 10 years of operation.

**Figure 2 materials-15-05796-f002:**
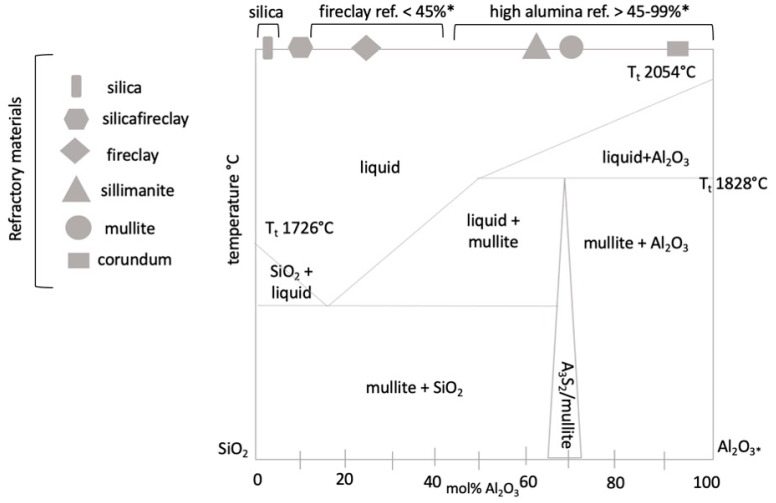
Categorization of basic refractory materials in binary diagram of SiO_2_-Al_2_O_3_. Note: * the amount of Al_2_O_3_.

**Figure 3 materials-15-05796-f003:**
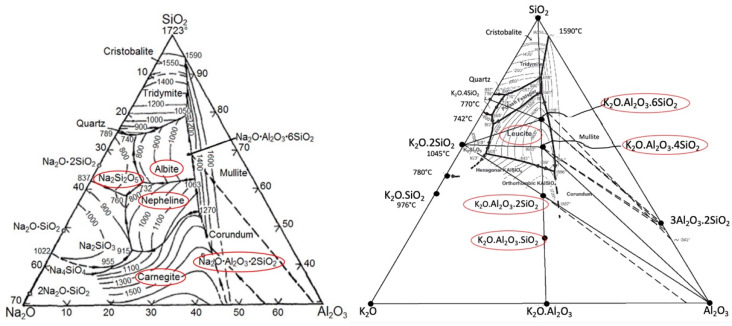
Ternary diagram of Na_2_O-SiO_2_-Al_2_O_3_ [13] and K_2_O-Al_2_O_3_-SiO_2_ [14], marking the individual phases formed during alkaline corrosion.

**Figure 4 materials-15-05796-f004:**
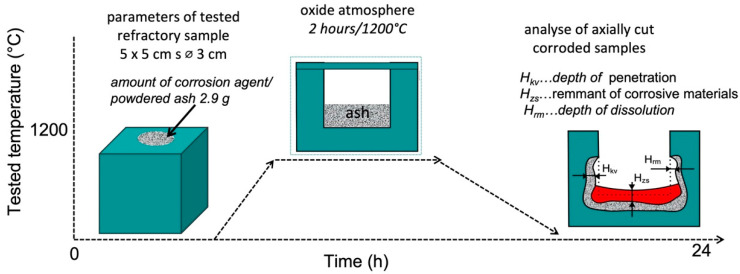
Schematic diagram of the crucible test of refractory materials.

**Figure 5 materials-15-05796-f005:**
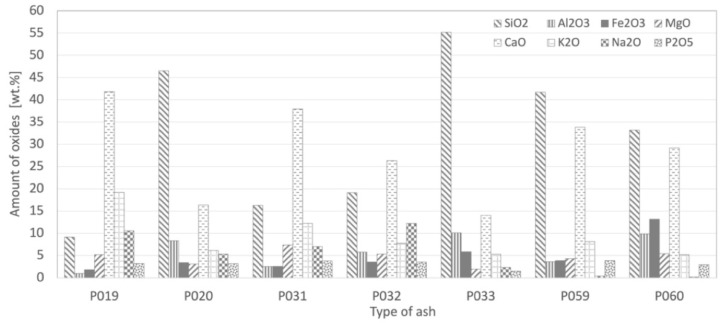
Concentrations of major elements in ash after wood biomass combustion.

**Figure 6 materials-15-05796-f006:**
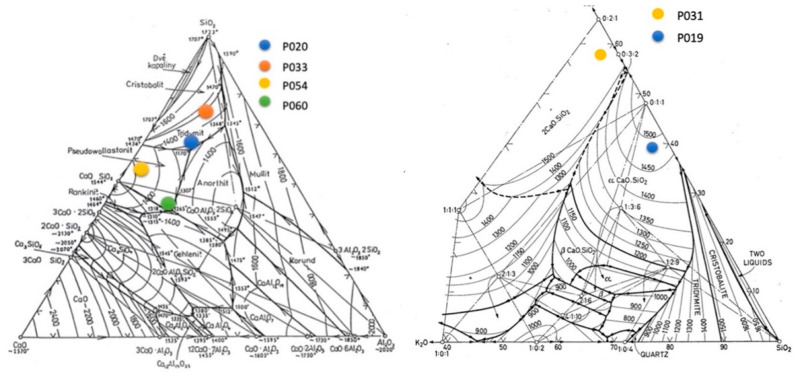
Approximate position of ash types P_020_, P_033_, P_059_, P_060_, in ternary diagram CaO-Al_2_O_3_-SiO_2_ [27] and P_031_ and P_019_ in ternary diagram K_2_O-Al_2_O_3_-SiO_2_ [28].

**Figure 7 materials-15-05796-f007:**
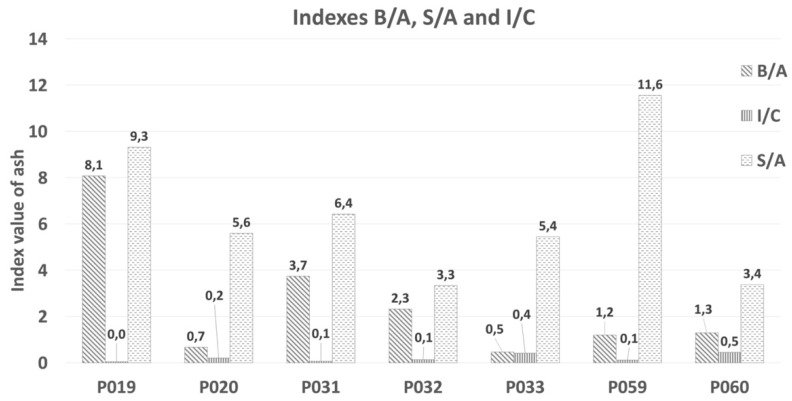
Indices B/A, S/A, and I/C for individual ash samples.

**Figure 8 materials-15-05796-f008:**
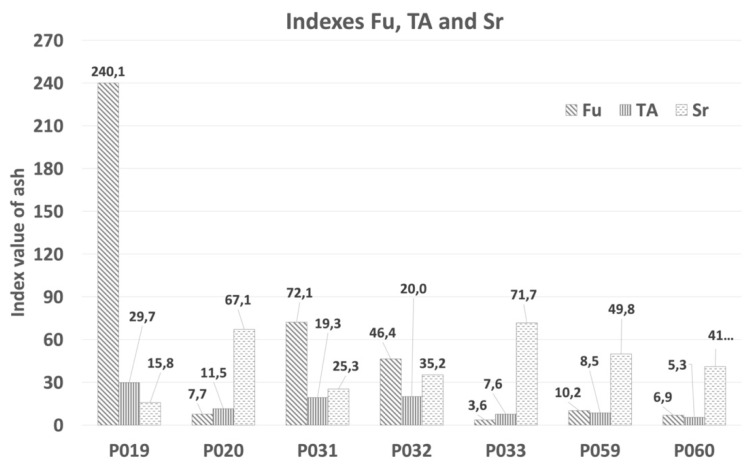
Indices for Fu, TA, and Sr for individual ash samples.

**Figure 9 materials-15-05796-f009:**
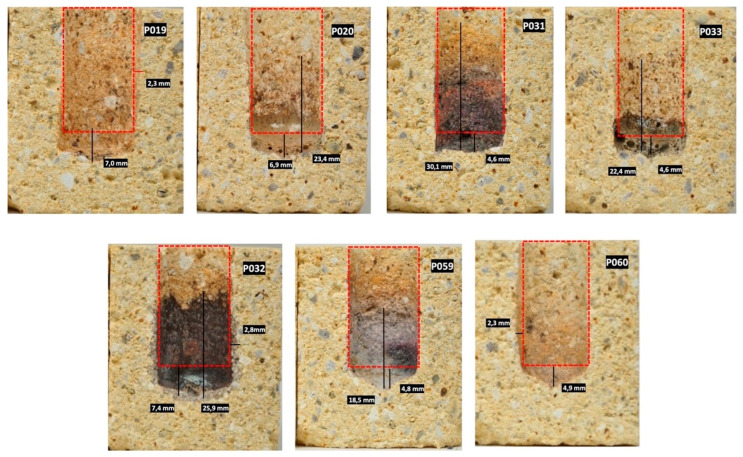
Corrosion of STV refractory materials using different types of wood ash and defined parameters of crucible test: 2.9 g ash/2 h/1200 °C.

**Figure 10 materials-15-05796-f010:**
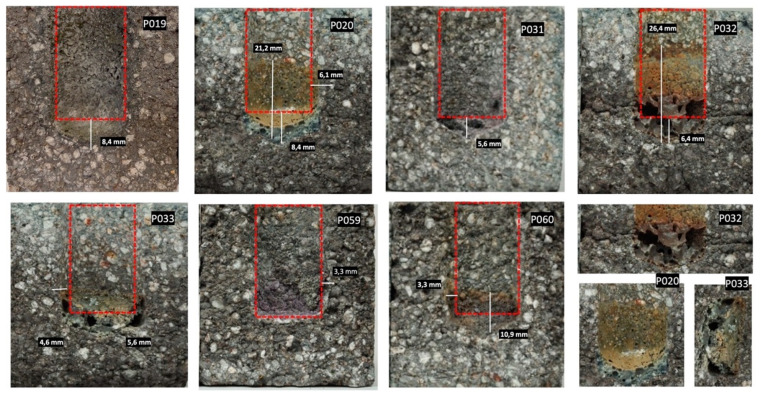
Corrosion of ARS60N quality refractory material with different types of wood ash with defined parameters of crucible test; 2.9 g ash/2 h/1200 °C.

**Table 1 materials-15-05796-t001:** Ash characterization indices [10,21,22,23,24].

Index	Equation	Tendency Slagging/Fouling			
		Low	Middle	High	Ex. High
SiO_2_ (%)	-	<20	20–25	>25	
Cl (%)	-	<0.2	0.2–0.3	0.3–0.5	>0.5
B/A	$\frac{B}{A}=\frac{{\mathrm{Fe}}_{2}O_{3}+\mathrm{CaO}+\mathrm{MgO}+{\mathrm{Na}}_{2}O+K_{2}O}{{\mathrm{SiO}}_{2}+{\mathrm{Al}}_{2}O_{3}+{\mathrm{TiO}}_{2}}$	<0.5	0.5–1	1–1.75	>1.75
S/A	$S/A=\frac{{\mathrm{SiO}}_{2}}{{\mathrm{Al}}_{2}O_{3}}$	<0.31	-	0.3–3	-
I/C	$\frac{I}{C}=\frac{{\mathrm{Fe}}_{2}O_{3}}{\mathrm{CaO}}$	<0.31	0.3–3	>3	-
Fu	$Fu=\frac{B}{A}\cdot \left({\mathrm{Na}}_{2}O+K_{2}O\right)$	<0.6	0.6–40	>40	-
TA	$TA={\mathrm{Na}}_{2}O+K_{2}O$	<0.3	0.3 < TA < 0.4	>0.4	-
Sr	$Sr=\frac{{\mathrm{SiO}}_{2}}{{\mathrm{SiO}}_{2}+{\mathrm{Fe}}_{2}O_{3}+\mathrm{CaO}+\mathrm{MgO}}\cdot 100$	>72	65–72	<65	

**Table 2 materials-15-05796-t002:** Characterization of wood ash used for experiment.

Type of Wood Biomass	Disposal Method	Labeled
Spruce pellets	combustion	P_019_
Woodchips	combustion	P_020_
Woodchips	combustion	P_031_
Woodchips, woodbark, sawdust, pellets, scraps	combustion	P_032_
Woodchips, woodbark, sawdust, pellets, scraps	combustion	P_033_
Woodchips	gassification	P_059,_ P_060_
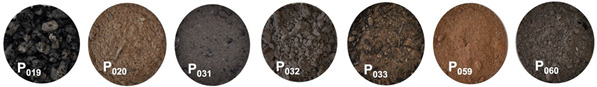

**Table 3 materials-15-05796-t003:** Chemical composition and properties of refractory materials.

Oxides wt.%	STV	ARS60N
SiO_2_	53.5	28.40
Al_2_O_3_	40.5	46.60
TiO_2_	2.1	-
Fe_2_O_3_	2.1	0.88
CaO	0.3	0.2
MgO	0.3	0.27
K_2_O + Na_2_O	0.8 + 0.2	0.5
SiC	-	13.2
Bulk density (kg/m^3^)	2150	2700
Apparent porosity (%)	18.0	15
Cold crushing strength (MPa)	30	70
Refractory qualities under load (RUL) T0.5 (°C)	1360	>1500

**Table 4 materials-15-05796-t004:** Alkali test classification after internal regulation of P-D Refractories CZ [26].

Class	Classification		
	Corrosion	Infiltration	Cracks
A	not attacked	no corrosion and/or infiltration	No
B	slight attack	<6 mm corrosion and/or infiltration	No
C	distinctive attack	>7 mm corrosion and/or infiltration	Slight
D	severe attack	>9 mm corrosion and/or infiltration	large, clearly visible cracks

**Table 5 materials-15-05796-t005:** Phase composition of analyzed biomass ash samples.

Phase Composition	Labeled of Sample						
	P_019_	P_020_	P_031_	P_032_	P_033_	P_059_	P_060_
quartz (SiO_2_)	x	x		x	x	x	
calcite (CaCO_3_)	x	x	X	x	x	x	X
graphite C	x						
CaO	x	x	X	x			
magnesite (MgCO_3_)			X				
MgO			X	x			
anorthite (CaAl_2_Si_2_O_8_)		x					
microcline (KAlSi_3_O_8_)		x			x	x	x
arcanite (K_2_SO_4_)			X				
anhydrite (CaSO_4_)				x			
anorthoclase					x		
leucite (KAlSi_2_O_6_)						x	x
orthoclase (KAlSi_3_O_8_)						x	
sylvite (KCl)							x
portlandite Ca(OH)_2_						x	x
hematite (Fe_2_O_3_)						x	x
mullite (Al_4.59_Si_1.41_O_0.97_)							x
analcime (NaAlSi_2_O_6_)							x

**Table 6 materials-15-05796-t006:** Calculation of slagging and fouling indices for individual ash types.

Ash	Index							
	SiO_2_ (%)	Cl (%)	B/A	S/A	I/C	Fu	TA	Sr
P_019_	9.1 ^l^	0.21 ^s^	8.1 ^ex^	9.3 ^h^	0.0 ^l^	240.1 ^h^	29.7 ^h^	15.8 ^h^
P_020_	46.5 ^h^	0.41 ^h^	0.7 ^m^	5.6 ^h^	0.2 ^l^	7.7 ^m^	11.5 ^h^	67.1 ^m^
P_031_	16.2 ^l^	0.16 ^l^	3.7 ^ex^	6.4 ^h^	0.1 ^l^	72.1 ^h^	19.3 ^h^	25.3 ^h^
P_032_	19.1 ^l^	1.74 ^ex^	2.3 ^ex^	3.3 ^h^	0.1 ^l^	46.4 ^h^	20.0 ^h^	35.2 ^h^
P_033_	55.1 ^h^	0.10 ^l^	0.5 ^l^	5.4 ^h^	0.4 ^m^	3.6 ^m^	7.6 ^h^	71.7 ^m^
P_059_	41.7 ^h^	-	1.2 ^h^	11.6 ^h^	0.1 ^l^	10.2 ^m^	8.0^5 h^	49.8 ^h^
P_060_	33.1 ^h^	-	1.3 ^h^	3.4 ^h^	0.5 ^m^	6.9 ^m^	5.3 ^h^	41.0 ^h^

Note: X ^l^: low value; X ^m^: middle value; X ^h^: high value; X ^ex^: extreme value.

**Table 7 materials-15-05796-t007:** Evaluation of STV refractory material corrosion at 1200 °C.

Quality	STV						
Ash	P_019_	P_020_	P_031_	P_032_	P_033_	P_059_	P_060_
Cracks	no	no	no	no	no	no	no
ČSN P CEN/TS 15418 [25]	U	U	A	A	LA	LA	U
PD Refractories CZ [26]	B	B	B	C	B	B	B

Note: Evaluation of crucible test according to [25] U: unaffected/no visible attack: −; LA: lightly attacked/minor attack: +; A attacked/clearly attacked: ++; C: corroded/completely corroded: +++. Evaluation of crucible test according to [26] A: not attacked; B: slight attack/<6 mm corrosion and/or infiltration/no cracks: +; C: distinctive attack/>7 mm corrosion and/or infiltration/slight cracks: ++; D: severe attack corrosion/>9 mm corrosion and/or infiltration/large cracks, visible disruption: +++.

**Table 8 materials-15-05796-t008:** Evaluation corrosion of refractory materials ARS60N at temperature 1200 °C.

Quality	ARS60N						
Ash	P_019_	P_020_	P_031_	P_032_	P_033_	P_059_	P_060_
Cracks	No	no	no	small	no	no	no
ČSN P CEN/TS 15418 [25]	LA	A	U	A	LA	U	LA
PD Refractories CZ [26]	B	C	B	C	B	B	B

Note: Evaluation of crucible test according to [25] U: unaffected/no visible attack: −; LA: light attacked/minor attack: +; A: attacked/clearly attacked: ++; C: corroded/completely corroded: +++. Evaluation of crucible test according to [26] A: not attacked; B: slight attack/<6 mm corrosion and/or infiltration/no cracks: +; C: distinctive attack/>7 mm corrosion and/or infiltration/slight cracks: ++; D: severe attack corrosion/>9 mm corrosion and/or infiltration/large cracks, visible disruption: +++.

## Data Availability

The data presented in this study are available on request from the corresponding author.

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
