# Peer review of "Corrosive Effect of Wood Ash Produced by Biomass Combustion on Refractory Materials in a Binary Al–Si System"

_materials, 2022, doi:10.3390/ma15165796_

Round 1

Reviewer 1 Report

- The paper deals with the current issue of corrosion of refractory linings of power plants burning biomass, or municipal waste. Due to the wide selection of biofuels, it is necessary to verify their corrosion effect to ensure smooth and safe operation of energy equipment.
- The main problem is the corrosion of refractory linings and the heat of exchangeable elements,
- The high content of alkali (Na and K) in the ashes causes the so-called "alkali burshing", degradation of the lining.
- In the paper, the corrosion effect on two types of high-aluminum refractory materials (STV) and with the addition of SiC (ARS60N) is experimentally monitored. A static crucible corrosion test was used, and the evaluation was done according to ČSN P CEN/TS 15418, or methods of Plibrico. Ash chemical composition analyzes using ED-XRF and mineralogical composition (XRPD) were used for ash evaluation.
- The slagability and stickiness of the used bio-ash was assessed based on the chemical composition using indices for the characterization of coal ash.

Additions and topics for authors:
- On the basis of which the crucible corrosion test temperature of 1200°C and the exposure time of 2 hours were chosen.
- For the evaluation of the corrosion tests, it would be advisable to use the analyzes of post-mortem samples (if possible) of the slag and refractory lining from the reaction zone.
- The thermal characteristics of the ashes (index of slagging and sticking) based on their chemical composition is sufficiently convincing, perhaps it would be appropriate to use monitoring of the fusibility of the ashes according to STN ISO 540,

Conclusion:
- The submitted contribution is original based on own experiments, using appropriate experimental and evaluation methodology.
- The results are processed clearly and comprehensibly.
- The obtained conclusions can be fully applied in both scientific and industrial practice.

a few little things:

line 258 ...5° min-1

line 348 ...the temperature selection does not match the assertions in lines 344-346 ?

line 381 ....ash P010 ???

line 405 I suggest: ... to attack ash at temperature 1200°C.

Author Response

REVIEWER 1

Dear reviewer, thank you very much for your comment, spent time and positive reaction on the topic. All English language recommendations have been corrected as recommended. English language was editing to quality for acceptable to print.

Thank you for commnet and and below are the answers.

Additions and topics for authors:
- On the basis of which the crucible corrosion test temperature of 1200°C and the exposure time of 2 hours were chosen.
- For the evaluation of the corrosion tests, it would be advisable to use the analyzes of post-mortem samples (if possible) of the slag and refractory lining from the reaction zone.
- The thermal characteristics of the ashes (index of slagging and sticking) based on their chemical composition is sufficiently convincing, perhaps it would be appropriate to use monitoring of the fusibility of the ashes according to STN ISO 540,

Testing temperature of 1200 °C is defined based on previous experience from other experiments. In our previous articles, we commented results for 1400 °C.Considering the chemical composition of the tested ash and the characteristics of the refractory materials, 1200 °C proved to be the optimum temperature.At the same time we believe that this temperature should not be exceeded too much in operation conditions.

Post-mortem analysis is a good comment. For the time being, we hav not evaluated this. In the presented article, we focused on laboratory testing and the way different types of ash react to selected refractory materials. This research is being continued and we will be happy to include post-mortem analysis into our program.

It will definitely be useful to harmonize measured data and it will be necessary to work with an international ISO norm. The evaluated article is part of more extensive research. STN ISO 540 based procedures will be gradually incorporated into the experiment.

Conclusion:
- The submitted contribution is original based on own experiments, using appropriate experimental and evaluation methodology.
- The results are processed clearly and comprehensibly.
- The obtained conclusions can be fully applied in both scientific and industrial practice.

a few little things:

line 258 ...5° min-1 /corrected

line 348 ...the temperature selection does not match the assertions in lines 344-346 ?/corrected

line 381 ....ash P010 ???/corrected

line 405 I suggest: ... to attack ash at temperature 1200 °C/corrected

Hana Ovčačíková and research teams.

Reviewer 2 Report

The authors in the manuscript “Effect of Corrosion by Wood Ash from Biomass Combustion on Refractory Materials in Binary System A-S” have attempted to analyse the effect of corrosion on 2 different types of composition. The paper is well written and has potential to get accepted. However, following comments can be incorporated.

1.      Tittle needs to be slightly modified.  

2.      Introduction is very lengthy. It needs to be reduced.

3.      Figure 1 and 2 needs modification. In figure 1 a, b and c are already mentioned. Figure 2 has 1, 2 and 3 which needs to be modified.

4.      Conclusions needs to be mentioned in the point wise which can be better for reader experience.    

Author Response

REVIEWER 2

Dear reviewer, thank you very much for your comment, spent time and positive reaction on the topic. All English language recommendations have been corrected as recommended. English language was editing to quality for acceptable to print.

Hana Ovčačíková and research teams.

The paper is well written and has potential to get accepted. However, following comments can be incorporated. 

Thank you for commnet and and below are the answers.

  1. Tittle needs to be slightly modified /corrected
  2. Introduction is very lengthy. It needs to be reduced./The text was reduced.
  3. Figure 1 and 2 needs modification. In figure 1 a, b and c are already mentioned. Figure 2 has 1, 2 and 3 which needs to be modified.

Recommendation of Reviewer was accepted and the point 1 and 2 were modified a treatment. The revision are marked in the text.  

  1. Conclusions needs to be mentioned in the point wise which can be better for reader experience../

It is true that the conclusion should be clear. I agree with that as the author. The conclusion was therefore given in single points.

 Hana Ovčačíková and research teams
